# New Insights into the Pro-Inflammatory and Osteoclastogenic Profile of Circulating Monocytes in Osteoarthritis Patients

**DOI:** 10.3390/ijms25031710

**Published:** 2024-01-30

**Authors:** Paloma Guillem-Llobat, Marta Marín, Matthieu Rouleau, Antonio Silvestre, Claudine Blin-Wakkach, María Luisa Ferrándiz, María Isabel Guillén, Lidia Ibáñez

**Affiliations:** 1Department of Biomedical Science, Cardenal Herrera-CEU University, 46115 Valencia, Spain; paloma.guillemllobat@uchceu.es; 2Department of Pharmacy, Cardenal Herrera-CEU University, 46115 Valencia, Spain; marta.marin1@uchceu.es; 3Laboratory of Molecular PhysioMedicine, UMR 7370, National Centre for Scientific Research, Côte d’Azur University, 06107 Nice, France; matthieu.rouleau@univ-cotedazur.fr (M.R.); claudine.blin@univ-cotedazur.fr (C.B.-W.); 4Service of Orthopedic Surgery and Traumatology, University Clinical Hospital, 46010 Valencia, Spain; antonio.silvestre@uv.es; 5Interuniversity Research Institute for Molecular Recognition and Technological Development (IDM), Polytechnic University of Valencia and University of Valencia, 46022 Valencia, Spain; luisa.ferrandiz@uv.es

**Keywords:** bone remodeling, monocyte, osteoclast progenitor, osteoarthritis, inflammation

## Abstract

Osteoarthritis (OA) is a degenerative condition of the articular cartilage with chronic low-grade inflammation. Monocytes have a fundamental role in the progression of OA, given their implication in inflammatory responses and their capacity to differentiate into bone-resorbing osteoclasts (OCLs). This observational–experimental study attempted to better understand the molecular pathogenesis of OA through the examination of osteoclast progenitor (OCP) cells from both OA patients and healthy individuals (25 OA patients and healthy samples). The expression of osteoclastogenic and inflammatory genes was analyzed using RT-PCR. The OA monocytes expressed significantly higher levels of CD16, CD115, TLR2, Mincle, Dentin-1, and CCR2 mRNAs. Moreover, a flow cytometry analysis showed a significantly higher surface expression of the CD16 and CD115 receptors in OA vs. healthy monocytes, as well as a difference in the distribution of monocyte subsets. Additionally, the OA monocytes showed a greater osteoclast differentiation capacity and an enhanced response to an inflammatory stimulus. The results of this study demonstrate the existence of significant differences between the OCPs of OA patients and those of healthy subjects. These differences could contribute to a greater understanding of the molecular pathogenesis of OA and to the identification of new biomarkers and potential drug targets for OA.

## 1. Introduction

Among the numerous hallmarks of aging, two important factors that hold the potential to serve as biomarkers have been described: bone marrow dysfunction and underlying non-infectious inflammation [1]. As a result of the combination of these two factors, some pathologies are exacerbated, and others arise. For example, the progressive loss of the trabecular bone structure unbalances the production of immune system cells, which, in turn, affects bone homeostasis. Human skeletal aging is caused by a gradual loss of bone mass due to an excess of bone resorption versus the formation of new bone. It has been shown that, in humans, the increase in osteoclastogenesis is dependent on the age of the individual, since the regulatory mechanisms are altered [2]. Accordingly, the chronic production of inflammatory mediators without an infectious reason is related to an advanced age. In fact, these mediators are considered markers of functional alteration and fragility, and of bone pathologies such as osteoporosis and osteoarthritis (OA) [3,4,5].

In the degenerative joint disease of OA, which is more common in older people, the progressive mechanical degradation of the articular cartilage occurs, affecting the subchondral bone and the entire joint cavity [6]. Consequently, both the alteration in bone metabolism and chronic inflammation are increased. This represents a worldwide therapeutic challenge with a high economic cost. Interestingly, until recently, it was thought that the only cause of OA was the mechanical degradation of cartilage. However, today, it is known that low-grade, chronic, non-infectious inflammation is a particularly key factor in the development and chronicity of this pathology [3,7]. Numerous cells of the joint cavity, such as chondrocytes and synovial membrane cells, and subchondral bone cells, such as osteoclasts (OCLs), osteoclast progenitors (OCPs), osteoblasts, etc., are involved in this pathology [8]. Thus, OA is a multifactorial inflammatory pathology with a non-autoimmune origin that affects the joints.

The subchondral bone is located deep in the articular cartilage, and is in direct contact with the cartilage and trabecular bone located below the subchondral plate [9]. It has been shown that there is an alteration in subchondral bone turnover during the evolution of OA. From the initial events of the pathology, the subchondral plate thickens due to the presence of calcified cartilage, and the trabecular bone shows reduced intertrabecular spaces. Bone formation is maintained in more advanced OA, which results in sclerosis [10]. Different CD45^+^ cell populations derived from hematopoietic cells, including monocytes, macrophages, and OCPs, have been identified in the subchondral area of cartilage. Among the CD45^+^ cells, more than 80% were identified as CD14^+^/CD68^−^ monocytes and some were identified as double-positive CD14^+^/CD68^+^ macrophages. A discrete subpopulation of OCPs CD45^+^/HLA-DR^−^/CD115^+^ was identified in cells isolated from the bone marrow. Histochemical staining demonstrated the presence of multinucleated TRAP^+^ osteoclasts (OCLs) on the bone surface and mononuclear TRAP^+^ cells [11]. 

OCLs are the only cells that have a bone resorption capacity. OCLs are multinucleated giant cells formed by the fusion of myeloid progenitors or mature myeloid cells (monocytes, macrophages, and dendritic cells) [12]. Macrophage colony-stimulating factor (M-CSF) and receptor-activating nuclear factor κB ligand (RANKL), produced by osteoblasts (the bone-forming cells) and stromal cells, modulate this process, enhancing the viability, proliferation, and differentiation of OCP cells into mature, functionally active osteoclasts. The binding of M-CSF to its receptor, c-fms/CD115, induces the expression of RANK on the surface of OCPs. Concomitantly, the RANKL/RANK interaction induces the activation of various transcription factors in OCPs, such as NF-kB, c-Fos, and NFATc1, which regulate OCL fusion and maturation, as well as other specific markers such as tartrate-resistant acid phosphatase (TRAP), cathepsin K, the calcitonin receptor, and integrins. Bone resorption occurs when OCLs become polarized and adhere to the bone surface known as the “sealing zone” [13,14].

OCLs, in addition to their function in bone remodeling, play an important role in the modulation of immune responses [12,15,16,17]. Human OCLs express major histocompatibility complex class II and the co-stimulatory molecules necessary for T stimulation. Thus, OCLs can process and present antigens and activate CD4^+^ and CD8^+^ T cells [18,19,20]. Furthermore, it has been shown that OCLs can induce immunosuppressive T cells, regulatory T cells, or inflammatory T cells depending on their progenitors or their context (healthy or with chronic inflammation) [20,21].

The complex and controversial relationship that exists between OCLs and inflammatory cytokines, with the most relevant role in OA, has recently been reviewed [13]. Treg cells expressing RANKL can be activated by IL-1, inducing the differentiation of OCLs and the consequent bone loss. However, it has also been shown that IL-1 can block RANK/RANKL signaling in human OCPs, thereby inhibiting osteoclastogenesis. On the other hand, the cytokine TNF-α is known to facilitate OCL formation in a multifactorial manner. Thus, TNF activates the RANK/RANKL pathway synergistically with other cytokines, such as RANKL or IL-1, in addition to increasing the expression of RANK in OCPs and the production of RANKL by osteoblasts and stromal cells. Independently of RANKL signaling, it can also directly induce the fusion and migration of OCPs, promoting OCL differentiation. Other inflammatory cytokines that participate in bone turnover in OA are IL-6 and IL-17. These cytokines can increase osteoclastogenesis independently of RANKL.

Monocytes are innate immune cells that play a significant role in the inflammatory response in OA [22]. They migrate to the subchondral area of the articular cartilage and differentiate into macrophages and OCLs [23]. Since monocytes are precursors of OCLs, their differentiation into this type of cell is particularly relevant for the study of the etiology and mechanism by which the pathogenesis of OA evolves. In addition, monocytes also produce inflammatory cytokines and enzymes that potentiate inflammation and, consequently, cartilage degradation [24]. Therefore, monocytes constitute a potential target for the diagnosis and treatment of this disease. The study of monocyte precursor populations of OCLs is complex, since there are various populations with distinctive characteristics and combinations of cell surface markers. The existence of three subpopulations of these cells is currently accepted, which show differences in their surface markers, phenotype, gene and protein expression, migration, cytokine secretion, and differentiation potential [25]. These subpopulations are known as classical (CD14^+^/CD16^−^), intermediate (CD14^+^/CD16^+^), and non-classical (CD14^−/+^CD16^++^) monocytes [26,27]. Although Sprangers et al. demonstrated that the three populations can differentiate in OCLs, this is still not fully clear, as reflected by the number of controversial studies addressing this issue [28].

Some studies have demonstrated the osteoclastogenic capacity of circulating OCPs in OA. On the one hand, it has been shown that peripheral blood mononuclear cells (PBMCs) from OA patients produce a greater number of OCLs with more bone resorption compared to healthy individuals, as well as showing a decreased apoptosis in OCLs [29]. Moreover, in these patients, there was no variation in the percentage of the total circulating CD14^+^. However, in another study, Loukov et al. found a lower number of monocytes in the peripheral blood of women with OA compared to healthy controls. On the other hand, these monocytes had an elevated expression of activation markers with a subsequent increase in the production of TNF and IL-1. It was also demonstrated that the CCR2 receptor was overexpressed in the population of classical and intermediate monocytes [30], which may explain the increase in the migration of OCPs to the joint and which may promote OCL differentiation and bone destruction, as described for other inflammatory bone erosion pathologies [31]. Also, it has been proven that there is an increase in the expression of genes related to the differentiation of OCLs in the PBMCs from patients with OA compared to those from healthy controls [32].

There are still few publications that have related the formation of OCLs exclusively to the pathology of OA, since most of the works show results indifferently from patients with rheumatoid arthritis and OA. Rheumatoid arthritis and OA are pathologies of a different origin and with different mechanisms, although the final joint and bone involvement is similar. Rheumatoid arthritis is a disease of autoimmune origin, where specific immune mechanisms, such as the activation of lymphocytes and the production of autoantibodies, are involved from the most initial stage of the pathology. On the contrary, OA in its origin is related to the innate immune response favored by the physiological aging of the individual [33]. The aging of the bone marrow alters the metabolism and cellular functionality of the immune system, causing OA, among other pathologies. On the other hand, there are no studies that have compared the possible cellular markers of OCPs circulating in the peripheral blood from OA patients with those from healthy individuals. Therefore, it would be difficult for us to find a possible therapeutic or diagnostic target. Taking these considerations into account, the objective of the present work was to study the possible differences in the expression of osteoclastogenic and inflammatory markers of OCPs from the peripheral blood monocyte populations of patients diagnosed with OA compared to those of healthy individuals. We also studied whether these markers reflect a greater osteoclastogenic and inflammatory capacity compared to healthy OCPs.

## 2. Results

### 2.1. Characteristics of OA Patients

Twenty-five patients undergoing a knee replacement were included in the study. The age range was between 56 and 83 years, with a mean ± SD age of 72.8 ± 7.1 years, and the group comprised 80% women and 20% men. The degree of OA was evaluated with the Kellgren–Lawrence scale, resulting in 17 patients with grade IV OA, 5 patients with grade III OA, and 1 patient with grade I OA (Table 1).

### 2.2. Expression of Osteoclastogenic and Inflammatory Markers of OA Monocytes

Monocytes have been determined to play an essential role in the development of OA lesions, both by contributing to the inflammatory microenvironment and by differentiating into bone-degrading osteoclasts [32]. For this reason, PBMCs were analyzed using flow cytometry to determine whether there were any significant differences in the percentage of these cells in the samples from healthy donors and OA patients. As shown in Figure 1, the percentage of monocytes in the myeloid cell population of each sample was found to be comparable in OA patients and healthy donors, showing no statistically significant differences (84.6 ± 9.1 vs. 84.1 ± 8.1).

The expression of genes such as CD16 and CD115 has been associated with the capacity of monocytes to differentiate into osteoclasts and, therefore, to promote bone degradation [14,34]. On the other hand, the induction of an inflammatory process through the activation of monocytes can take place via the upregulation of PAMP (pathogen-associated molecular pattern) receptors such as TLR2, Mincle, or Dectin-1 [35,36,37]. Additionally, inflammation can be enhanced by an increase in monocyte recruitment due to the overexpression of chemokine receptors such as CCR2 [38]. Therefore, to determine whether the expression of these genes was different in circulating monocytes isolated from OA patients compared to healthy donors, the levels of mRNAs were assessed using RT-PCR. After the isolation of CD14^+^ cells from the peripheral blood, our results showed that there was a significant increase in the expression of CD16, CD115, TLR2, Mincle, Dectin-1, and CCR2 in OA patients vs. healthy donors (Figure 2A).

Considering that the biological action of these receptors depends on their surface expression, a flow cytometry analysis was carried out to determine the expression levels of CD16, CD115, and TLR2 in OA and healthy CD14^+^ cells. In the case of the CD16 and CD115 receptors, the results confirmed those found in the RT-PCR assays. However, in the case of TLR2 expression, no significant differences were found between the surface expression of this toll-like receptor in the monocytes from OA patients and healthy donors (Figure 2B).

### 2.3. Peripheral Blood Monocyte Subsets in OA Patients

Previous studies have identified three subsets of circulating monocytes that may possess different capacities regarding their differentiation potential as well as their ability to promote inflammatory responses [39,40,41]. Considering this, the percentage of classical (CD14^++^CD16^neg^), intermediate (CD14^++^CD16^+^), and non-classical (CD14^+^CD16^++^) monocytes was studied in the blood samples from the OA patients and healthy donors. As shown in Figure 3, there were no significant differences in the intermediate and non-classical monocyte populations. The classical subset of monocytes turned out to be the predominant one when comparing both CD16^+^ subpopulations. In addition, it should be noted that this CD14^++^ population was significantly increased in OA patients compared to healthy donors (81.06 ± 6.55 vs. 73.30 ± 6.14; *p* = 0.0089).

Next, we went on to study the expression of CD115 and TLR2 in each of the individual subsets of monocytes identified using flow cytometry. Only in the non-classical monocyte population, the percentage of cells expressing CD115 was found to be significantly increased in the OA patients compared to the control subjects (97.76 ± 2.09 vs. 90.23 ± 9.16; *p* = 0.0259; Figure 4). The individual analysis of the other two populations did not show differences for the mentioned receptors.

### 2.4. Osteoclastogenesis Potential of Monocytes from OA Patients

Using PCR and cytometry, our results show that the OA CD14^+^ monocytes had a higher expression of CD115 compared to the control monocytes. Considering the relevance of bone resorption in the progression of OA, these data are interesting, since CD115 is an indispensable factor for the differentiation of OCPs to OCLs [12,42]. Therefore, we examined the differentiation capacity of circulating monocytes isolated from the peripheral blood of patients with OA and healthy donors.

TRAP-positive cells with three or more nuclei were scored as osteoclasts, and the levels of expression of cathepsin K and RANK were used to check the adequate differentiation of the monocyte’s cultures into active OCLs. In this sense, the differentiation protocol of the OCLs was validated, showing a strong expression of cathepsin K and RANK in the OCLs compared to the CD14 monocytes (Figure 5). Furthermore, we observed a significant increase in the TRAP^+^ multinucleated OCL formation in the cultures from OA patients compared to those from healthy controls (190 ± 70 OCLs per well in the healthy group vs. 343 ± 212 OCLs per well in the OA group, *p* = 0.0268; Figure 6A,B). These results suggest a greater potential for OA monocytes to differentiate into OCLs compared to those from healthy donors.

### 2.5. Inflammatory Potential of Monocytes from OA Patients

The inflammatory microenvironment is a determinant for the development and progression of lesions in OA patients [43]. Consequently, to clarify whether the increased levels of TLR2 and Dectin-1 mRNAs in the monocytes from OA patients could provide a predisposition for an individual to respond in a more pronounced way to pro-inflammatory stimuli, circulating monocytes were exposed to zymosan, a well-characterized ligand for both receptors [44,45]. 

After the stimulation of the OCLs, as described in the Materials and Methods, the IL-6 production was analyzed using ELISA. IL-6 is a cytokine closely related to bone metabolism and is one of the main inflammatory cytokines that enhance inflammation in OA [46]. No differences were observed in the basal production of this cytokine by the unstimulated cells of OA patients vs. controls (19.99 ± 6.8 pg/mL vs. 20.5 ± 19.2 pg/mL; Figure 4). As expected, the cells from both healthy subjects and OA patients produced IL-6 in response to zymosan treatment. However, interestingly, the monocytes isolated from the blood samples of OA patients showed a significantly greater production of IL-6 when stimulated, compared to the stimulated monocytes from healthy donors (2.2 *×* 10^4^ ± 0.9 *×* 10^4^ pg/mL vs. 1.3 *×* 10^4^ ± 0.9 *×* 10^4^ pg/mL; *p* = 0.0423; Figure 6C).

## 3. Discussion

In this work, we found interesting differences when comparing the peripheral blood monocyte populations of patients with OA and healthy individuals. In relation to the markers of osteogenesis and inflammation, in the monocytes of OA patients compared to control patients, there was a significant increase in the expression of mRNA for the Mincle, Dectin-1, TLR2, and CD115 receptors, accompanied by a greater expression on the cell surface of CD16 and CD115. On the other hand, although no differences were seen in the percentage of the total circulating monocytes between OA and healthy patients, we found that the majority of the population was CD14^+^+CD16^neg^, while the CD14+CD16^++^ population expressed more CD115 on its surface vs. control monocytes. We also showed that OA monocytes have a greater osteoclastogenic and inflammatory capacity compared to healthy ones.

In this study, we did not find any significant differences in the percentage of circulating monocytes in the blood of OA patients compared to healthy subjects. This agrees with a previous study that also focused on OA [29]. The analysis of the expression of genes and surface markers in the monocytes from OA and healthy controls that we showed in the present study revealed several relevant differences. Interestingly, our results demonstrated an increase in the levels of Mincle and Dectin-1 mRNAs in patients vs. controls. This is, to our knowledge, the first description of this differential expression between circulating monocytes obtained from OA patients and those obtained from healthy controls. The implications of the enhanced expression of these two receptors are still to be unraveled and appear highly relevant, due to their role as PAMP and DAMP (damage-associated molecular pattern) receptors and their participation in the modulation of inflammatory responses [35,36,47,48]. Intriguingly, previous work has shown no differences between the mRNA levels of Dectin-1 in the synovial tissue of OA patients vs. controls [49]. Moreover, in this work, the expression of the chemokine receptor CCR2 was significantly increased in OA monocytes compared to healthy ones. This finding is consistent with those of other authors, which supports the fact that circulating monocytes, in the context of OA inflammation, are recruited to the injured area [30,50].

Our results also showed an increase in the TLR2 mRNA expression in OA patients’ monocytes. However, no significant difference was observed in the TLR2 surface expression. This finding could indicate that monocytes in OA patients are somehow predisposed or primed to respond more rapidly or more intensely when exposed to the pro-inflammatory microenvironment found in the lesion area. Indeed, the production of IL-6 in response to zymosan was shown to be significantly increased in the monocytes isolated from OA patients compared to those isolated from healthy controls. This effect could be attributed to the higher expression of Dectin-1, the increased levels of TLR2 mRNA, or the coordinated participation of both receptors [37,51]. Nevertheless, circulating monocytes from patients seem to have a greater inflammatory potential that could be relevant for the progression of OA lesions.

On the other hand, we described, for the first time, a significant increase in CD115 expression in circulating monocytes isolated from the blood of OA patients compared to controls. Given the key role of this receptor in the induction of osteoclastogenesis and the survival of OCLs [14,52,53], this finding may have a significant impact on the understanding of the mechanisms involved in the progression of OA. Indeed, these results are, to our knowledge, the first pinpoint analysis of the differential expression of this receptor in OA patients vs. healthy subjects, since few previous studies have addressed this issue and, moreover, they mainly focused on the synovial tissue of rheumatoid arthritis, rather than the circulating monocytes of OA [54,55]. Therefore, this finding may constitute the basis for new research opportunities to identify the underlying pathological mechanisms and novel therapeutic targets.

In the case of CD16, our findings demonstrated a significant increase in both the mRNA and surface expression of this marker in the monocytes from OA patients compared to those from controls. These results are consistent with previous studies in which the surface expression of CD16 was analyzed using flow cytometry in OA monocytes [30]. Moreover, studies focusing on other inflammatory diseases, such as psoriatic arthritis, have also shown higher percentages of CD16^+^ monocytes in patients compared to healthy subjects [34].

Based on the expressions of CD14 and CD16, peripheral blood monocytes have been classified into three subpopulations with different surface markers and immune-related properties: classical, intermediate, and non-classical monocytes [39,40,41]. In a recent review, Tsai J. et al. pointed out that, in different inflammatory diseases such as rheumatoid arthritis, which have a different etiology compared to OA, the proportion of intermediate and non-classical monocytes varies in the synovial fluid and in the blood. For example, in patients with psoriatic arthritis or multiple myeloma, the population of CD16^+^ monocytes are increased in the blood, with a greater capacity to differentiate into OCLs compared to other monocyte subsets. Contrarily, other studies have proven that OCLs are mainly derived from the classical CD14^+^CD16^−^ monocyte population in healthy donors and patients with rheumatoid arthritis. It has also been described that CD14^+^CD16^+^ monocytes can respond to RANKL to produce TNFα and IL-1 but cannot differentiate into osteoclasts [25]. Our results show that the predominant subset of circulating monocytes was the one defined as classical, which is in line with previous work [39,56]. Interestingly, the percentage of these classical monocytes increased significantly in OA patients compared to controls. Furthermore, the individual analysis of the expression of surface markers for each of the monocyte subsets showed that the non-classical subpopulation in OA patients expressed significantly higher levels of CD115.

Regarding the distribution and shift of monocyte subpopulations during the progression of OA and other bone-related diseases, further clarification is needed, considering the different results that have been published by several authors. These controversial studies may account for different analytical approaches or may also be explained by the grade of OA and the clinical conditions of the patients that have been included in the studies. Also, the classification of the monocyte subpopulations based on the expression of CD14 and CD16 has recently been questioned and may be reconsidered by including additional markers for these subsets [39,41,57]. This fact could definitively have an important impact on the design of future work, as well as on the interpretation of results. Hence, the standardization of methodological and analytical approaches will be essential for a reliable comparison of the significance of the different studies addressing the shift of monocyte subpopulations during pathology progression. Moreover, the use of certain drugs such as glucocorticoids has been shown to affect the distribution of the monocyte subpopulation, specifically by reducing the non-classical subset [58]. This fact may also explain the diversity in the results, given that patients with chronic inflammatory diseases are very commonly treated long-term with pain killers and anti-inflammatory drugs.

In this context, the increased number of classical monocytes and the enhanced expression of CD115 may predispose patients to developing bone loss, since classical monocytes have been associated with cells that transform into osteoclasts in rheumatoid arthritis by several authors [59,60,61], and the activation of CD115 by MCSF promotes a greater expression of RANK and, therefore, induces an intensified osteoclastogenic capacity. Our results show an increase in CD155 expression in the non-classical monocyte population of OA patients compared to controls. Non-classical monocytes are colloquially known as patrolling monocytes, since they are continually scanning blood vessels, quickly targeting damaged tissue [39]. Perhaps non-classical monocytes could be the first or most relevant monocytes to be recruited in the areas of OA lesions to differentiate into OCLs.

In another phase of this study, we verified the osteoclastogenic potential of OA monocytes compared to control monocytes by evaluating the expression of the CTSK and TNFRSF11A genes. Indeed, the results demonstrated that there was a greater capacity for differentiation in the OCLs derived from OA monocytes vs. those derived from controls. We also demonstrated in primary cultures of monocytes that there is indeed a greater capacity for the differentiation of OA monocytes compared to healthy ones. It should also be noted that the production of the cytokine IL-6 significantly increased in OA monocytes compared to control ones when the cells were incubated with the inflammatory agent zymosan. These results agree with a study by Durand M et al., although it was only tested in a physiological situation [29].

To improve the diagnosis and control of the evolution of OA, the early identification of symptoms is essential. However, this currently remains difficult in clinical practice for different reasons. OA in its initial stages normally presents with mild and non-specific symptoms and can often be confused with normal aging or fatigue. In addition, sometimes due to its symptoms, OA can be confused with some types of rheumatic disease, mainly rheumatoid arthritis in the early stage of the disease. Obviously also in the early phases, an imaging diagnosis does not have a sufficient sensitivity to detect minimal changes in the damaged joint and, furthermore, it does not discriminate OA from rheumatoid arthritis. It is very important to highlight that there are currently no specific biomarkers in the blood that can differentiate OA from other rheumatic diseases with joint involvement. In conclusion, our work was framed in this study in the context of a search for new cellular targets for a more sensitive and specific OA diagnosis. We believe that the characterization of OCPs can be an essential tool that facilitates the early diagnosis of this pathology. Our results open many possibilities to continue in this line of translational research for the diagnosis and therapy of OA. Despite the efforts in this regard, there is still much to do, since a major limiting factor that we encountered was the availability of samples from patients with a clear diagnosis of OA and from healthy individuals.

## 4. Materials and Methods

### 4.1. Human Sample Collection

In this work, an observational–experimental study was carried out with human monocytes obtained from peripheral blood in the period of 2019–2023. Human monocytes were isolated from buffy coats from healthy donors provided by the Blood Bank Valencia Transfusion Center. OA blood samples (10 mL) were obtained from 25 white patients (5 men and 20 woman, 72.8 ± 7.07 years (mean ± SD) diagnosed with advanced OA and undergoing a total knee arthroplasty at the Orthopedic Surgery and Traumatology Unit of the Clinical Hospital of Valencia, after they provided informed consent according to the Declaration of Helsinki. Patients with a diagnosis or suspicion of rheumatoid arthritis or other rheumatic pathology were excluded. The study was conducted in accordance with the Declaration of Helsinki and approved by the Ethics Committee for Biomedical Research of the CEU Cardenal Herrera University (code CEI19/009; date: 10 January 2023), and the Ethical Committee for Research with Medications of the Valencia University Clinical Hospital (protocol: version 3; 12 June 2023; code 2023/101; date: 30 June 2023). All the patients met the American College of Rheumatology criteria for symptomatic OA. The exclusion criteria included a history of inflammatory arthritis, an intra-articular corticosteroid injection (within 3 weeks of the surgery), and blood dyscrasias. Radiographs from the patients were evaluated to determine the Kellgren–Lawrence (KL) grading. Early-stage KOA was defined by KL grades I/II and late-stage was defined by grades III/IV. The experimental design was approved by the institutional ethical committees (University of Valencia and La Fe Polytechnic University Hospital, Valencia, Spain).

### 4.2. Isolation of Human PBMCs and CD14^+^ OCPs

Human PBMCs were isolated from either blood samples obtained from OA patients or buffy coats from healthy donors. The blood was centrifuged for 20 min at 1500 rcf in BD Vacutainer^®^ tubes (BD Bioscience, San Jose, CA, USA). After centrifugation, the white blood cells were collected, washed with phosphate-buffered saline (PBS), and counted. The resulting cells were used for a flow cytometry analysis or further purified using the positive magnetic selection of CD14^+^ cells. 

For CD14^+^ cell purification, the PBMCs were then incubated with CD14 magnetic microbeads (Miltenyi Biotec, Auburn, CA, USA) and isolated using a MiniMACS™ Separator and Starting Kit (Miltenyi Biotec, Auburn, CA, USA), following the manufacturer’s instructions. The purified CD14^+^ cells were then counted again to use them for cell culture experiments or a qRT-PCR analysis.

### 4.3. ARN Extraction and qPCR

For gene expression, the total cellular RNA was extracted with Trizol (Invitrogen, Carlsbad, CA, USA) from purified CD14 cells conserved in RNAlater (Thermo Fisher Scientific, Waltham, MA, USA) and reverse-transcribed with random primers and SuperScript II Reverse Transcriptase (Invitrogen, Thermo Fisher Scientific). The RNA was quantified using NanoDrop technology (Invitrogen). The amplification of cDNA was performed using a quantitative RT-PCR StepOnePlus Fast real-time PCR system (Applied Biosystems, Foster City, CA, USA) with the SensiFact SYBR Hi-Rox kit (BIOLINE, London, UK), using the following primers (Table 2):

The dissociation curve method was applied according to the manufacturer’s protocol (60 °C to 95 °C) to ensure the presence of a single specific PCR product. The quantification of mRNA expression relative to the 36B4 mRNA expression (used as a reference standard) was determined using the comparative delta-ct (2^−ΔCT^) method.

### 4.4. Flow Cytometry Analysis

The PBMCs were isolated as aforementioned, and aliquots of 2 *×* 10^6^ cells were incubated with the following antibodies in order to select the OCP populations of interest: APC-Cy7-conjugated CD14 (BD Bioscience; clone MφP9), FITC-conjugated CD16 (Miltenyi Biotec, Paris, France), PE-conjugated CD3 (BD Pharmingen, San Diego, CA, USA), PE-conjugated CD56 (Miltenyi), PE-conjugated CD19 (BD Bioscience), and PE-conjugated CD15 (BD Pharmingen). Additionally, for each experimental condition, the cells were incubated with one of the following markers of interest: APC-conjugated TLR2 (Molecular Probes), APC-conjugated CD115 (BD Pharmingen), APC-conjugated CLEC4E (R&D Systems, Minneapolis, MN, USA), or APC-conjugated TLR4 (Miltenyi). Appropriate controls were used to determine the optimal voltage settings and the electronic subtraction for the spectral fluorescence overlap correction. The samples were analyzed in a CytoFlex instrument (Beckman Coulter, Brea, CA, USA) and elaborated using the CytExpert software version 2.5.0.77 (Beckman Coulter, Inc.).

### 4.5. CD14^+^ Cell Culture and Stimulation

The CD14^+^ cells were maintained in Roswell Park Memorial Institute culture medium (RPMI; Gibco, Paisley, UK) supplemented with 1% penicillin/streptomycin and 10% fetal bovine serum (FBS; Gibco) at 37 °C in 5% CO_2_ and seeded at a density of 3.5 × 105 cells/well in 24-well plates. Following the 24 h rest period, the cells were treated with zymosan (10 μg/mL) for 24 h. The supernatants were collected and stored at −80 °C until use. Adherent cells were washed with PBS and detached with an accutase solution for 15 min at room temperature. The remaining cells were detached by repeated pipetting using PBS. The cells were transferred into 15 mL tubes and centrifuged at 300× *g* for 10 min at room temperature. The cell pellet was resuspended in 100 µL of RNAlater and stored at −18 °C until the analysis.

### 4.6. ELISA on the IL-6 Protein Production

The IL-6 amount in the cell culture supernatants was analyzed using a human IL-6 instant enzyme-linked immunosorbent assay (ELISA) kit (Invitrogen, Thermo Fisher Scientific) according to the manufacturer’s instructions. The lower limit of detection was 0.92 pg/mL.

### 4.7. Osteoclast Differentiation and TRAP Staining

Purified CD14^+^ monocytes were incubated in 96-well plates (6 × 10^4^ cells/well) in MEMα (Gibco, Life Technologies, Carlsbad, CA, USA) supplemented with 10% fetal bovine serum (HyClone™-characterized FBS, Logan, UT, USA), 1% penicillin/streptomycin, 25 ng/mL of M-CSF (PeproTech, London, UK), and 50 ng/mL of RANKL (PeproTech), at 37 °C and 5% CO_2_. The medium was replaced every 3 days. The cells required 12–14 days to become multinucleated giant cells. Then, the cells were stained using a tartrate-resistant acid phosphatase (TRAP) staining kit (Sigma-Aldrich, St. Louis, MO, USA), according to the manufacturer’s protocol. In each well, the number of TRAP-positive cells with ≥3 nuclei were counted under a light microscope (Leica DMi1 inverted microscope, Wetzlar, Germany) and imaged.

### 4.8. Statistical Analysis

The data are expressed as the mean and standard deviation (mean ± SD). The normality of the distribution was tested using the Shapiro–Wilk normality test. The differences between the OA patients and the controls were examined using the Mann–Whitney U test for non-normally distributed variables. For normal distributions, the *t*-test was employed. The GraphPad Prism (v10.0.3, GraphPad Software Inc., San Diego, CA, USA) software was used for the analysis and the graphical design of the data. The significance level was set at *p* < 0.05.

## 5. Conclusions

Our study reveals distinct molecular and phenotypic differences in peripheral blood monocytes of osteoarthritis (OA) patients, suggesting heightened inflammatory and osteoclastogenic potential. The identified shift in monocyte subsets and increased osteoclastogenic capacity emphasize the intricate immune dynamics in OA. Thus, our results advocate for exploring monocyte markers, such as CD115 and TLR2, as potential biomarkers for sensitive OA diagnosis and therapeutic targets.

## Figures and Tables

**Figure 1 ijms-25-01710-f001:**
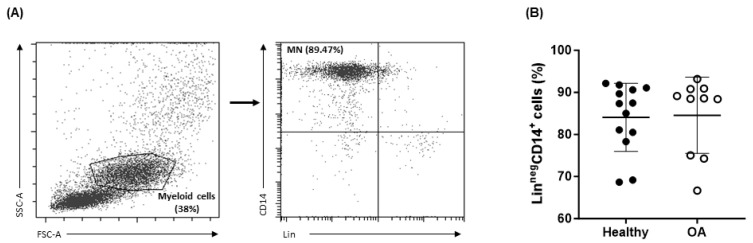
Percentage of circulating CD14^+^ monocytes in peripheral blood from OA patients and healthy donors. Representative dot plots of the gating strategy used for the identification of the monocyte population after doublet removal are shown in (**A**). Lin- (CD56, CD3, CD19, CD15) CD14^+^ cells were considered monocytes (MNs). Percentage of MNs found from the total myeloid cells in each of the samples is shown in (**B**). Data are presented as mean ± SD.

**Figure 2 ijms-25-01710-f002:**
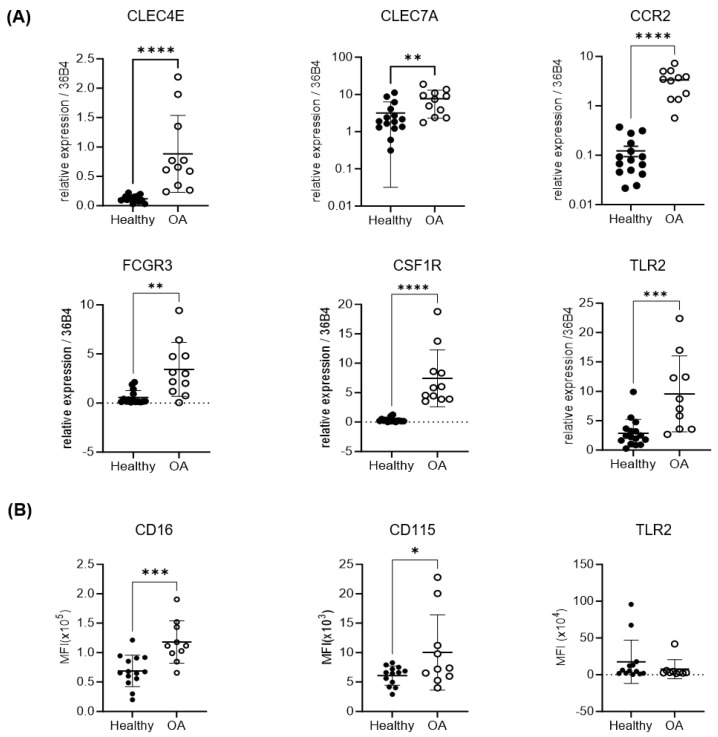
Osteoclastogenic and inflammatory marker analysis in monocytes. (**A**) Gene expression analysis in monocytes from peripheral blood of OA patients and healthy donors. CLEC4E = Mincle gene; CLEC7A = Dectin 1 gene; CCR2 = CCR2 gene; FCGR3 = CD16 gene; CSF1R = CD115 gene; and TLR2 = TLR2 gene. (**B**) Flow cytometry analysis of surface expression of receptors CD16, CD115, and TLR2 in monocytes from peripheral blood of OA patients and healthy donors. MFI = mean fluorescence intensity. *n* = 12−15 samples. Data are presented as mean ± SD. * *p* < 0.05; ** *p* < 0.01; *** *p* < 0.001; and **** *p* < 0.0001.

**Figure 3 ijms-25-01710-f003:**
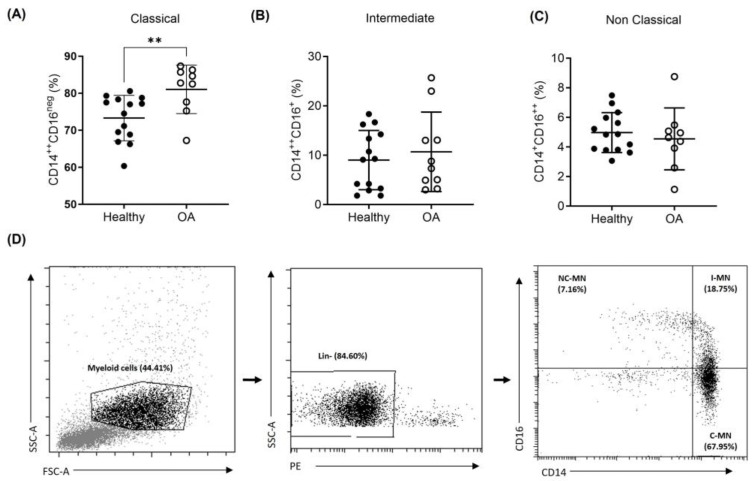
Monocyte subsets in peripheral blood from OA patients and healthy donors. Percentage of classical monocytes (CD14^++^/CD16^−^) (**A**); intermediate monocytes (CD14^++^/CD16^+^) (**B**); and non-classical monocytes (CD14^+^/CD16^++^) (**C**). A representative dot plot of the gating strategy for the identification of the monocyte subsets is shown in (**D**). *n* = 10–14 samples. Data are presented as mean ± SD. ** *p* < 0.01.

**Figure 4 ijms-25-01710-f004:**
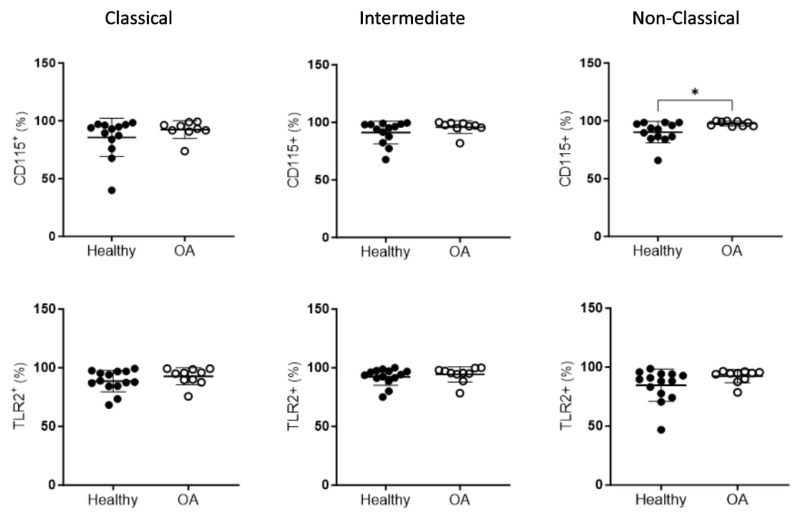
Expression of receptors CD115 and TLR2 in individual monocyte subsets. OA and healthy samples from peripheral blood were analyzed using flow cytometry. *n* = 9–14 samples. Data are presented as mean ± SD. * *p* < 0.05.

**Figure 5 ijms-25-01710-f005:**
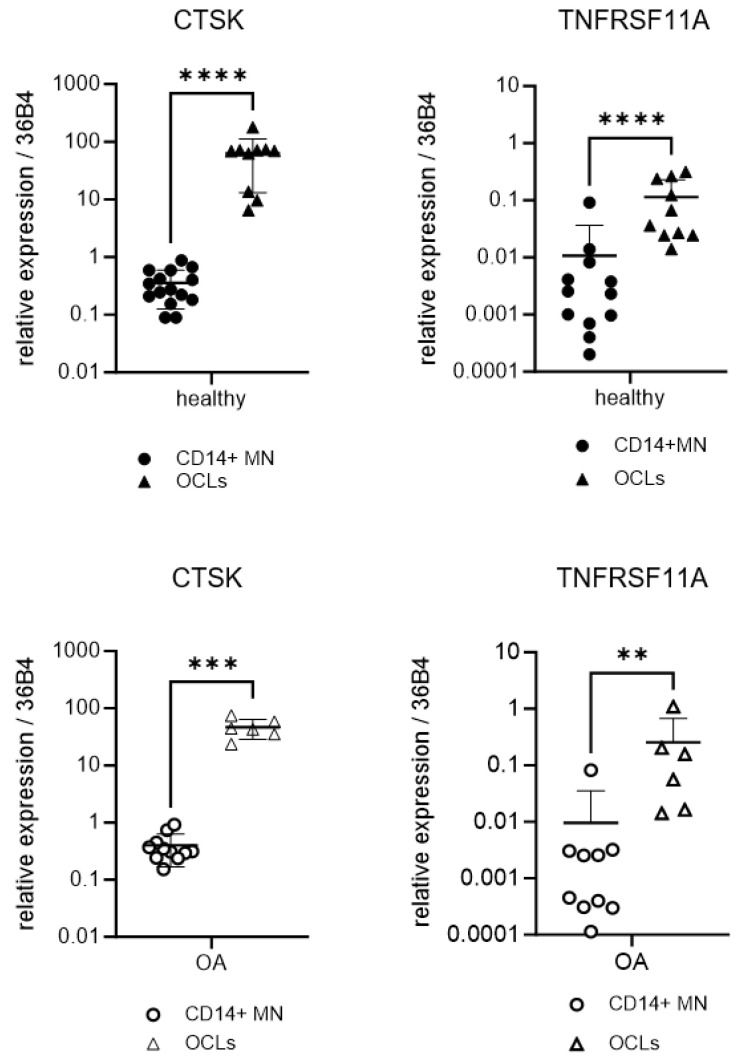
CTSK and TNRFSF11A gene expression in monocytes and OCLs of OA patients and healthy donors. CTSK = cathepsin K gene; TNRFSF11A = RANK gene. Circles represent CD14^+^ cells; triangles represent OCLs. *n* = 5–15 samples. Data are presented as mean *±* SD. ** *p* < 0.01; *** *p* < 0.001; and **** *p* < 0.0001.

**Figure 6 ijms-25-01710-f006:**
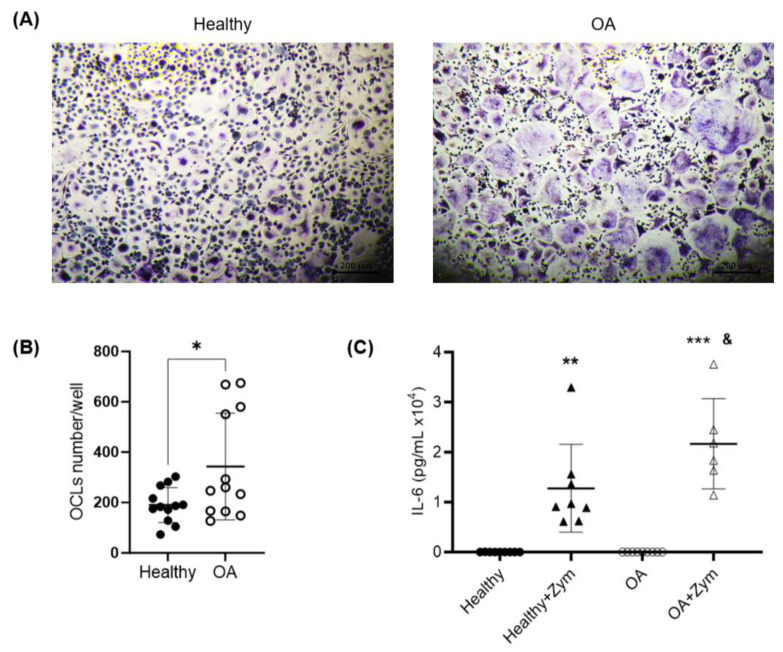
Osteoclastogenic and inflammatory capacity of OA monocytes. (**A**) Representative TRAP-stained OCL cultures from CD14^+^ monocytes isolated from OA and health patients. TRAP^+^ multinucleated cells (>3 nuclei) were considered OCLs (magnification ×4). (**B**) Results of OCL counting per well in the cultures from OA patients and healthy donors. (**C**) Quantification of IL-6 production by monocytes from OA patients and healthy donors, unstimulated or stimulated with zymosan. *n* = 6–12. Data are presented as mean ± SD. * *p* < 0.05 vs. healthy; ** *p* < 0.01 and *** *p* < 0.001 vs. unstimulated control; and ^&^
*p* < 0.05 OA vs. healthy + zymosan.

**Table 1 ijms-25-01710-t001:** Characteristics of OA patients.

PATIENT	SEX	AGE (Years)	KL GRADE	ANALGESIA	CRP (mg/L)
1	Female	73	IV	INITIAL.MINOR	2.1
2	Female	80	IV	NO	4
3	Female	72	IV	NO	4.5
4	Female	82	III	SECOND STEP	<1
5	Female	74	II	NO	2
6	Female	71	IV	NO	2.5
7	Male	69	III	NO	<1
8	Female	76	IV	NO	<5
9	Female	74	IV	OCASSIONAL	4
10	Female	73	II	DAILY	<1
11	Female	78	IV	NO	<1
12	Female	69	IV	OCASSIONAL	<1
13	Female	83	IV	OCASSIONAL	4
14	Female	63	IV	NO	<1
15	Female	66	IV	NO	5
16	Female	81	III	DAILY	<5
17	Female	70	IV	DAILY	<5
18	Male	56	IV	OCASSIONAL	<5
19	Male	58	III	NO	<5
20	Female	78	IV	NO	<1
21	Male	83	III	NO	<5
22	Male	70	IV	NO	<1
23	Female	70	IV	NO	<1
24	Female	74	IV	INITIAL.MINOR	5
25	Female	77	IV	INITIAL.MINOR	<5

KL GRADE: Numerical value of OA according to Kellgren–Lawrence scale. CRP: C-reactive protein; values ≤ 5 mg/L were considered non-significant.

**Table 2 ijms-25-01710-t002:** Primer sequences used for RT-PCR.

36B4:	Fwd-TGCATCAGTACCCCATTCTATCAT Rv-AGGCAGATGGATCAGCCAAGA
CLEC4E:	Fwd-CTGAAACACAATGCACAGAGAGA Rv-AAAGATGCGAAATGTCACAACAC
CLEC7A:	Fwd-GGAAGCAACACATTGGAGAATGG Rv-CTTTGGTAGGAGTCACACTGTC
CCR2:	Fwd-GATCTGCTTTTTCTTATTACTCTCCCA Rv-TCCGCCAAAATAACCGATGT
FCGR3:	Fwd-GTCACTGTCCCAAGTTGCTAAG Rv-TCCTTCCTGTGTGCTTGTGG
CSF1R:	Fwd-CCAGCAGCGTTGATGTTAACTTT Rv-CGGCATGTTGGAAATCTACTTG
TLR2:	Fwd-TTGTGACCGCAATGGTATCTG Rv-TGTTGGACAGGTCAAGGCTTT
CTSK:	Fwd-TGAGGCTTCTCTTGGTGTCCATAC Rv-AAAGGGTGTCATTACTGCGGG
TNFRSF11A:	Fwd-ACCTTGCCTTGCAGGCTACTT Rv-CAAACCGCATCGGATTTCTC

## Data Availability

The data generated during this research have been deposited at the Cardenal Herrera-CEU University, in the Research Laboratory of the research group, and are available to anyone who may request them.

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
