# Peer review of "New Insights into the Pro-Inflammatory and Osteoclastogenic Profile of Circulating Monocytes in Osteoarthritis Patients"

_ijms, 2024, doi:10.3390/ijms25031710_

Round 1
Reviewer 1 Report
Comments and Suggestions for Authors
This study entitled “New insights into the pro-inflammatory and osteoclastogenic profile of circulating monocytes in osteoarthritis patients” seems to have been generally well executed and written. Furthermore, I believe that this paper will be of great interest to the readers. However, I have a few remarks that require authors attention, and a few suggestions to further improve the quality of this work.
Title
Please add the type of article in your title.
Abstract
Please state the type of study you have performed, and the number of enrolled and ultimately analyzed patients.
Keywords
Consider some additional MeSH keywords to readers easier identify your research.
1.Introduction
Please state a more clear and more specific the aim of your study.
Please state the hypothesis of your study at the end of Introduction.
Remove the findings of your study from the Introduction.
4.Materials and Methods
4.1. Human samples collection
Please add in the first sentence an information what type of study you have performed, and in which time period.
Please state the number of Ethical approval and the date when the approval was gained.
Did you register your study (e.g., ClinicalTrials.gov). If yes, please state the number and the date of registration.
4.8. Statistical analysis
Why the sample size calculation was not performed?
Results
Please state exact P values not just p<0.05 (e.g., 97.76±2.09 vs 90.23±9.16; p<0.05; Figure 4).
Discussion
Please begin Discussion with the main findings of your study.
Please state the limitations of your study at the end of Discussion.
Reviewer 2 Report
Comments and Suggestions for Authors
This is a very well written manuscript. However, I have some comments
- authors should include more references to previous studies to establish the background and relevance of the research in the introduction section
- there is an incomplete description of sample collection methods - authors should provide detailed information about sample collection, including the number of samples, demographic details, and exclusion criteria.
- Ensure that interpretations in the discussion section are directly supported by the data
- Expand the discussion to explore how the findings contribute to the current clinical practice
- please rework the conclusion section - it is overly broad and unspecific conclusions.
Reviewer 3 Report
Comments and Suggestions for Authors
The manuscript submitted by Guillem-Llobat et al. describes the pro-inflammatory and osteoclastogenic profile of circulating monocytes in osteoarthritis patients. The authors conducted a comprehensive characterization of pro-inflammatory and osteoclastogenic markers in osteoarthritis patients to contribute new insights for identifying potential biomarkers and drug targets for OA. While the study offers interesting insights, several issues need attention before recommending it for publication.
Major Concerns to be Addressed:
1. In Figure 1, the authors should provide a more detailed presentation of the gating strategy. Specifically, did the authors gate the singlets only for downstream analysis
2. In Figure 2, the authors should use gene symbols rather than a mixture of gene and protein names in the figure (e.g., CLEC4E for mincle, CLEC7A for dectin 1). Additionally, the authors should explain/show the full term for the abbreviation of MFI (either "Median" or "Mean" Fluorescence Intensity) since it is unclear to the readers.
3. In Figure 3, the gating is not clear. The authors claimed that non-classical monocytes are CD14+CD16++, however, it seems that CD14- has also been included. If the authors pre-gated the CD14+ with proper controls/other PBMCs, please show the gating strategy. Additionally, the threshold of CD16 appears peculiar. It seems like the authors split a very tight population into two parts. What is the reason/evidence for setting the threshold at that position (CD16 Y-axis)?
4. In Figure 4, please ensure that the Y-axis is on the same scale to facilitate comparability among different types of monocytes.
5. Is there any difference in CD115 and TLR2 expression between healthy and OA patients in general, regardless of monocyte type?
6. Ensure consistency throughout the entire manuscript with the decimal separator, either using a comma or a dot. Especially in line 276, the mixed use in one sentence is not acceptable.
Minor Concerns to be Addressed:
1. Please maintain consistency throughout the manuscript when referring to figures. For example:
In line 192, the authors used "Figure 2_panelA"
In line 277, the authors used "Figure 6C"
2. In the figure captions, if the authors are using capital letters "A/B/C...", they should also use capital letters in the figure captions rather than lowercase letters "a/b/c."
Comments on the Quality of English Language
Minor editing of English language required
Round 2
Reviewer 3 Report
Comments and Suggestions for Authors
Overall, the revised version improved the quality of the manuscript.
However, there are still some issue left from my previous comments.
Ensure consistency throughout the entire manuscript with the decimal separator, either using a comma or a dot. Although authors claimed that dots have been kept as decimal separator. I can still see lots of comma used as decimal separator in both the highlighted text and original text.
Line 240
Line 427
Line 434
Comments on the Quality of English Language
N/A.
Author Response
Following the reviewer's instructions, we have changed what he indicated. In addition, we have reviewed the possible existence of other errors.